# Bridging Transformers and RWKV: Towards Efficient Multimodal Video Understanding

## Abstract

Transformer-based Multimodal Large Language Models (MLLMs) struggle to process hour-long video inputs due to the quadratic computational complexity of causal self-attention, leading to prohibitively high computational costs during both training and inference. Existing token compression approaches reduce the number of video tokens, but often suffer from significant information loss and remain inefficient for extremely long sequences. In this work, we propose a hybrid RWKV-Transformer model that distills transformer layers into linear RNNs by reusing their attention projection weights, guided by a progressive distillation strategy. Without any token reduction, when replacing about $25\%$ of standard Transformer layers with RWKV modules improves throughput by $20\%$ compared to the original Transformer model, while matching its performance on multiple video understanding benchmarks such as Video-MME and MLVU, and even outperforming it on **VNBench** and **LVBench**, with average scores of $\mathbf{74.0}\%$ and $\mathbf{46.8}\%$, respectively.

## 1 Introduction

Inspired by LLMs (Achiam et al., 2023; Touvron et al., 2023), MLLMs (Li et al., 2023a; Zhu et al., 2023; Li et al., 2025; Liu et al., 2023) have advanced multimodal understanding, especially for images. Recent Transformer-based MLLMs excel at high-res image (Laurençon et al., 2023; Liu et al., 2024b; Wang et al., 2024b) and interleaved image-text tasks (Jiang et al., 2024; Li et al., 2024a;b). However, their quadratic self-attention complexity limits long-context processing, hindering effective long-video understanding.

A key challenge for MLLMs in long-video processing is encoding each frame into many visual tokens, incurring high compute/memory costs. Prior work mitigates this by reducing token counts: some use Q-Former (Li et al., 2023a) for compression (Fei et al., 2024; Li et al., 2023b;c; Zhang et al., 2023); others chunk and adaptively compress tokens (Shen et al., 2024; Shu et al., 2025). Yet, aggressive compression loses critical visual details in ultra-long videos, and self-attention's quadratic cost remains a bottleneck. An alternative approach uses cross-attention to let text embeddings interact with visual features (Alayrac et al., 2022; Li et al., 2025; Liu et al., 2024a), but limits interaction frequency and lacks persistent visual context within the LLM.

To mitigate Transformer complexity, efficient architectures like Mamba (Gu & Dao, 2023; Dao & Gu, 2021) (SSM-based) and RWKV (Peng et al., 2023; 2024; 2025) (linear RNN) replace softmax attention with hardware-friendly operations, boosting throughput. Recent works (Xu et al., 2025; Li et al., 2024c; Lu et al., 2024) apply them to accelerate video understanding. However, as shown in Chen et al. (2024b); Zhang et al. (2024a), linear RNNs suffer history decay beyond training length, degrading long-context performance. Hybrid models like LongLLaVA (Wang et al., 2024d) and Vamba (Ren et al., 2025) combine Mamba with Transformers to retain modeling strength while gaining efficiency, but require scratch training and suffer instability on long sequences (Yu & Erichson, 2025), often underperforming pure Transformers despite speed gains.

To improve throughput without token reduction and minimize training cost, we propose a hybrid RWKV-Transformer architecture for video MLLMs. Specifically: (1) We map pre-trained attention weights directly to RWKV, enabling full parameter reuse without retraining; (2) We employ cross-attention to help RWKV capture global context, mitigating its inherent history decay.

Experimental results show that replacing Transformer layers with RWKV modules increases throughput by up to nearly $\mathbf{2}\times$. In particular, replacing $\mathbf{25}\%$ of the Transformer layers with RWKV achieves performance comparable to the original model on multiple video benchmarks and even exceeds the original Transformer-based model by $\mathbf{0.6}\%$ on VNBench (Zhao et al., 2024b) and by $\mathbf{1.5}\%$ on LVBench (Li et al., 2023b).

Our summarized contributions are as follows:

- We are the first to propose a hybrid RWKV-Transformer architecture for video multimodal large models, enabling significant inference acceleration without token reduction or retraining.
- We introduce a parameter remapping strategy that directly transfers pre-trained attention weights to RWKV modules and design a progressive distillation strategy, preserving learned representations while minimizing training cost. And we leverage cross-attention to assist RWKV in capturing global contextual information.
- We demonstrate that replacing Transformer layers with RWKV modules increases throughput by up to nearly $\mathbf{2}\times$. In particular, replacing $\mathbf{25}\%$ of the Transformer layers with RWKV modules achieves performance comparable to the original model across multiple video benchmarks, and even surpasses the original Transformer-based model on VNBench and LVBench.

## 2 RELATED WORKS

**Video MLLMs.** Existing Transformer-based video MLLMs (Li et al., 2023b; Maaz et al., 2023; Zhang et al., 2023; Wang et al., 2024b; Cheng et al., 2024) append per-frame visual features, but quadratic attention causes slow inference. Most mitigate this via token compression: heuristic pooling (Xu et al., 2024a; Zhang et al., 2024b; Xu et al., 2024b), similarity-based merging (Chai et al., 2024; Shen et al., 2024), or trainable modules like Q-Former (Li et al., 2023a) and cross-attention (Liu et al., 2024e). However, aggressive compression often discards fine-grained details, degrading performance.

**RNN-based Large Language Models.** While Transformer-based LLMs (Achiam et al., 2023; Touvron et al., 2023; Team, 2024) excel in NLP, their quadratic complexity motivates renewed interest in RNNs (Feng et al., 2024; Li et al., 2018; Peng et al., 2025; Roemmele & Gordon, 2018; Shen et al., 2018), which offer constant per-token cost. Linear attention (Katharopoulos et al., 2020) reduces complexity to $\mathcal{O}(N)$ via kernel approximations, with variants (Chai & Xu, 2025; Gu & Dao, 2023; Dao & Gu, 2021; Peng et al., 2023; 2024; 2025; De et al., 2024) achieving strong results. RWKV (Peng et al., 2023; 2024; 2025) uniquely combines Transformer-style training with RNN-style inference. However, as Chen et al. (2024b) shows, such models struggle to extrapolate beyond training length. Our hybrid RWKV-Transformer preserves efficiency while enhancing long-context modeling, without sacrificing parallelism or scalability.

## 3 PRELIMINARIES

Before formally introducing our efficient hybrid architecture for video understanding, we first define its two key components:

### 3.1 ATTENTION MECHANISM

Let $\boldsymbol{x} \in \mathbb{R}^{N \times C}$ denote a sequence of $N$ features with dimension $C$. Single head **Softmax attention** Vaswani et al. (2017), also known as dot-product attention, can be written as:

$$\boldsymbol{Q} = \boldsymbol{x}\boldsymbol{W}_Q, \ \boldsymbol{K} = \boldsymbol{x}\boldsymbol{W}_K, \ \boldsymbol{V} = \boldsymbol{x}\boldsymbol{W}_V, \ \ \boldsymbol{y}_i = \sum_{j=1}^{N} \frac{\exp\left(\boldsymbol{Q}_i\boldsymbol{K}_j^\top/\sqrt{d}\right)}{\sum_{j=1}^{N}\exp\left(\boldsymbol{Q}_i\boldsymbol{K}_j^\top/\sqrt{d}\right)}\boldsymbol{V}_j \quad (1)$$

where $\boldsymbol{W}_Q, \boldsymbol{W}_K \in \mathbb{R}^{C \times d}$, $\boldsymbol{W}_V \in \mathbb{R}^{C \times C}$ denote projection matrices, $\boldsymbol{Q}, \boldsymbol{K} \in \mathbb{R}^{N \times d}$, $\boldsymbol{V} \in \mathbb{R}^{N \times C}$ represent query/key/value matrices, and $\boldsymbol{Q}_i, \boldsymbol{K}_i \in \mathbb{R}^{1 \times d}, \boldsymbol{V}_i \in \mathbb{R}^{1 \times C}$ are individual query/key/value tokens. Softmax attention computes the similarities between each query-key pair, leading to $\mathcal{O}(N^2)$

complexity. Therefore, it incurs unbearable computational cost in long-sequence modeling scenarios.

**Linear attention** (Katharopoulos et al., 2020), another attention paradigm, is proposed to effectively address this problem by reducing the computation complexity to $\mathcal{O}(N)$. Specifically, linear attention replaces the non-linear Softmax function with linear normalization, and adopts an additional kernel function $\phi$ in $\boldsymbol{Q}$ and $\boldsymbol{K}$:

$$\boldsymbol{Q} = \phi(\boldsymbol{x}\boldsymbol{W}_Q), \quad \boldsymbol{K} = \phi(\boldsymbol{x}\boldsymbol{W}_K), \quad \boldsymbol{V} = \boldsymbol{x}\boldsymbol{W}_V,$$

$$\boldsymbol{y}_i = \sum_{j=1}^{N} \frac{\boldsymbol{Q}_i \boldsymbol{K}_j^\top}{\sum_{j=1}^{N} \boldsymbol{Q}_i \boldsymbol{K}_j^\top} \boldsymbol{V}_j = \frac{\boldsymbol{Q}_i \left( \sum_{j=1}^{N} \boldsymbol{K}_j^\top \boldsymbol{V}_j \right)}{\boldsymbol{Q}_i \left( \sum_{j=1}^{N} \boldsymbol{K}_j^\top \right)}. \tag{2}$$

This enables the rearrangement of the computation order from $\left( \boldsymbol{Q}\boldsymbol{K}^\top \right) \boldsymbol{V}$ to $\boldsymbol{Q} \left( \boldsymbol{K}^\top \boldsymbol{V} \right)$ based on the associative property of matrix multiplication, thus reducing computation complexity to $\mathcal{O}(N)$.

Equation (2) presents linear attention featuring a complete sequence context, where each query gathers information from all keys and values. When applied to autoregressive models, linear attention can be constrained to limit the context of the $i$-th token to preceding tokens only (where $j \leq i$). This causal variant of linear attention is expressed as:

$$\boldsymbol{y}_i = \frac{\boldsymbol{Q}_i \left( \sum_{j=1}^{i} \boldsymbol{K}_j^\top \boldsymbol{V}_j \right)}{\boldsymbol{Q}_i \left( \sum_{j=1}^{i} \boldsymbol{K}_j^\top \right)} = \frac{\boldsymbol{Q}_i \boldsymbol{S}_i}{\boldsymbol{Q}_i \boldsymbol{Z}_i}, \quad \boldsymbol{S}_i = \sum_{j=1}^{i} \boldsymbol{K}_j^\top \boldsymbol{V}_j, \ \ \boldsymbol{Z}_i = \sum_{j=1}^{i} \boldsymbol{K}_j^\top. \tag{3}$$

This results in a recurrent linear attention form:

$$\boldsymbol{S}_i = \boldsymbol{S}_{i-1} + \boldsymbol{K}_i^\top \boldsymbol{V}_i, \ \ \boldsymbol{Z}_i = \boldsymbol{Z}_{i-1} + \boldsymbol{K}_i^\top, \ \ \boldsymbol{y}_i = \boldsymbol{Q}_i \boldsymbol{S}_i / \boldsymbol{Q}_i \boldsymbol{Z}_i. \tag{4}$$

### 3.2 RWKV

**RWKV**(Peng et al., 2023) combines the parallelizable training efficiency of Transformers with the sequential inference capability of RNNs. Its recurrent mechanism only references the immediately preceding token, enabling unbounded sequence length during inference without increasing memory consumption. The core architecture of RWKV-4 computes a weighted sum of past values, modulated by an accept vector, to efficiently facilitate information flow across time steps. we use $D$ to denote the model dimension and use $h$ to denote the number of heads. This can be expressed as:

$$\boldsymbol{wkv}_t = \frac{\sum_{i=1}^{t-1} \exp\big(-(t-1-i)\boldsymbol{w} + \boldsymbol{k}_i\big) \odot \boldsymbol{v}_i + \exp(\boldsymbol{u} + \boldsymbol{k}_t) \odot \boldsymbol{v}_t}{\sum_{i=1}^{t-1} \exp\big(-(t-1-i)\boldsymbol{w} + \boldsymbol{k}_i\big) + \exp(\boldsymbol{u} + \boldsymbol{k}_t)}, \tag{5}$$

$$\boldsymbol{y}_t = \boldsymbol{r}_t \cdot \boldsymbol{wkv}_t.$$

where $t$ is the current time index, $i \in \{1, \ldots, t-1\}$ indexes historical steps, $\boldsymbol{w} \in \mathbb{R}^{D/h}$ is the learnable decay rate, $\boldsymbol{u} \in \mathbb{R}^{D/h}$ is a learnable vector that emphasizes attention to the current token, $\boldsymbol{k}_i, \boldsymbol{k}_t \in \mathbb{R}^{D/h}$ are key vectors, $\boldsymbol{v}_i, \boldsymbol{v}_t$ are value vectors, and $\boldsymbol{r}_t \in \mathbb{R}^{D/h}$ is the receptance vector that acts as the receiver of past information.

The $\boldsymbol{wkv}_t$ attention calculation can alternatively be written in a recurrent form:

$$\boldsymbol{wkv}_t = \boldsymbol{S}_{t-1} + \mathrm{diag}(\boldsymbol{u}) \cdot \boldsymbol{k}_t^T \cdot \boldsymbol{v}_t,$$

$$\boldsymbol{S}_t = \mathrm{diag}(\boldsymbol{w}) \cdot \boldsymbol{S}_{t-1} + \boldsymbol{k}_t^T \cdot \boldsymbol{v}_t. \tag{6}$$

For computational convenience, we set $\mathrm{diag}(\boldsymbol{w}) = \mathrm{diag}(\boldsymbol{u}) = \mathbf{I}$, where $\mathbf{I}$ denotes the identity matrix. The above equations then simplify to:

$$\boldsymbol{wkv}_t = \boldsymbol{S}_t = \boldsymbol{S}_{t-1} + \boldsymbol{k}_t^T \cdot \boldsymbol{v}_t,$$

$$\boldsymbol{y}_t = \boldsymbol{r}_t \cdot \boldsymbol{S}_t. \tag{7}$$

Building upon RWKV-4, RWKV-5 employs matrix-valued hidden states to enhance model expressivity, while RWKV-6 introduces data-dependent token shifting and linear interpolation between input tokens to further improve sequence modeling capability. RWKV-7 enhances sequence modeling by dynamically updating its recurrent state based on contextual interactions between keys and values. Crucially, RWKV maintains linear time complexity $\mathcal{O}(N)$ and constant memory usage during inference, making it highly efficient for long-sequence modeling.

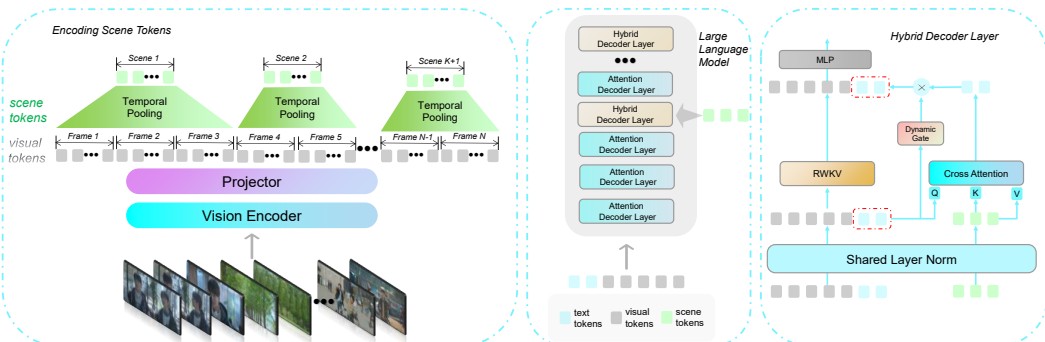

Figure 1: Hybrid architecture overview. **Left**: video frames are encoded into visual tokens, segmented into scenes via inter-frame similarity, and pooled into scene tokens. **Middle**: visual, text, and scene tokens are fed into the hybrid LLM; scene tokens are only used in hybrid decoder layers. **Right**: visual and text tokens pass through RWKV; scene tokens interact with text via cross-attention, and outputs are fused via dynamic gating.

## 4 HYBRID RWKV-TRANSFORMER ARCHITECTURE

As shown in Figure 1, our hybrid architecture integrates a visual encoder, projector, and LLM. Video features are segmented by adjacent-token similarity, then temporally pooled into scene tokens—serving as global anchors to mitigate RWKV's history decay. Meanwhile, original features are preserved. Hybrid layers inject cross-attention into pre-trained decoders, enabling text embeddings to retrieve fine-grained details. A dynamic gating mechanism further modulates per-token visual absorption.

### 4.1 ENCODING SCENE TOKENS

Video content often contains substantial redundancy, as consecutive frames within the same scene typically convey highly similar visual information. Processing every frame independently may lead to redundant computation and potential overfitting. To mitigate this, conventional approaches(Shi et al., 2025; Jiang et al., 2025) apply fixed-interval time pooling to reduce redundancy. However, such methods often merge frames across scene boundaries, which can severely compromise the semantic quality of the pooled representations. In contrast, our method first computes pairwise similarity scores between tokens of adjacent frames:

$$s_t = \frac{1}{\text{Sim}(\boldsymbol{V}_t, \boldsymbol{V}_{t+1})} \tag{8}$$

where $\boldsymbol{V}_t$ denotes the visual token sequence at frame $t$, and $\text{Sim}(\cdot, \cdot)$ is a cosine similarity function.

We then select the top-$k$ similarity scores $s_t$ that the similarity is less than the threshold $\tau$, and record their corresponding frame indices as scene boundaries, denoted by the set $\mathcal{B}$:

$$\mathcal{B} = \left\{ t \,\middle|\, s_t \in \text{TopK}\left(\{s_i\}_{i=1}^{T-1}\right) \text{ and } s_t > \frac{1}{\tau} \right\} \tag{9}$$

where $T$ is the total number of frames, $s_t$ is the similarity score between frame $t$ and $t+1$, and $\text{TopK}(\cdot)$ returns the indices of the $k$ highest values.

The video is then segmented into $|\mathcal{B}|+1$ scenes, and each scene is independently pooled to generate compact scene tokens for global context modeling. Let the sorted boundary indices be denoted as $\mathcal{B} = \{b_1, b_2, \ldots, b_{|\mathcal{B}|}\}$ with $b_1 < b_2 < \cdots < b_{|\mathcal{B}|}$. We define auxiliary boundaries $b_0 = 0$ and $b_{|\mathcal{B}|+1} = T$ for notational convenience. Then, for the $j$-th scene ($j = 1, 2, \ldots, |\mathcal{B}| + 1$), spanning frames from $b_{j-1} + 1$ to $b_j$, its scene token $\boldsymbol{V}_{s,j}$ is computed via temporal average pooling:

$$\boldsymbol{V}_{s,j} = \frac{1}{b_j - b_{j-1}} \sum_{t=b_{j-1}+1}^{b_j} \boldsymbol{V}_t \tag{10}$$

## 4.2 HYBRID DECODER LAYER

At the core of our hybrid architecture is the hybrid decoder layer, enabling text-scene interaction during LLM forward pass. As in Figure 1, it replaces self-attention with RWKV-v6-Finch (Peng et al., 2024) for efficiency. Cross-attention, parallel to RWKV, retrieves visual context from scene tokens and merges via skip connection. A dynamic gate modulates this output to reduce interference and stabilize training. We detail its computation below. Our model is built on Qwen2.5-VL (Bai et al., 2025) by replacing selected self-attention layers with hybrid layers.

**Cross-Attention.** The input to the hybrid decoder layer consists of hidden states and scene tokens. Inspired by mPLUG-Owl3 Ye et al. (2024), both inputs are first processed through a shared layer normalization layer. After normalization, the hidden states are processed in parallel by RWKV and cross-attention modules. All hidden states are fed into RWKV for sequential modeling. For cross-attention, however, the queries are restricted to text tokens only. We denote the query matrix as $\boldsymbol{Q}_t \in \mathbb{R}^{n \times d}$, where $n$ is the number of text tokens and $d$ is the hidden dimension (head dimension omitted for simplicity).

The cross-attention output $\boldsymbol{X}'$ is computed by projecting the text embeddings and scene tokens into query, key, and value spaces:

$$\boldsymbol{X}' = \mathrm{Attn}\left(\boldsymbol{W}_Q \boldsymbol{X}_t,\ \boldsymbol{W}_K \boldsymbol{V}_s,\ \boldsymbol{W}_V \boldsymbol{V}_s\right) \tag{11}$$

where Attn represents the multi-head cross attention. $\boldsymbol{X}_t \in \mathbb{R}^{n \times d}$ denotes the layer-normalized hidden states of text tokens, and $\boldsymbol{W}_Q, \boldsymbol{W}_K, \boldsymbol{W}_V \in \mathbb{R}^{d \times d}$ are learnable projection matrices for the cross-attention module.

**Dynamic gate.** While cross-attention output X enriches text tokens with visual context, it risks interfering with the pre-trained LLM. Conventional methods use a static, learnable scalar gate (Alayrac et al., 2022; Dubey et al., 2024), applying uniform weights regardless of token-specific relevance. Nevertheless, such a static gate applies uniform weights across all tokens. In practical applications, the requirement for visual context attention can vary substantially among tokens and input instructions, as demonstrated by heterogeneous attention patterns (Zhu et al., 2024). To overcome this constraint, Shi et al. (2025) introduce a dynamic gating mechanism incorporating a warm-up strategy. This architecture enables the model to determine, for each token individually, the proportion of cross-attention output to incorporate into each text token's embedding. In contrast to Shi et al. (2025), who route detailed tokens to cross-attention and coarse tokens to self-attention, where the loss of fine-grained visual information in self-attention impairs temporal reasoning and representation coherence, and then merge them before a shared output projection, our method feeds all visual tokens into RWKV while using scene tokens only for enhancement, and applies separate output projections followed by dynamic gating, thereby preserving fine-grained details and maintaining the specialized function of each component to enable better reuse of pre-trained weights.

Specifically, a single linear layer followed by a $\mathrm{tanh}$ activation is applied to the text embeddings to generate a dynamic gating vector $\boldsymbol{g}_d \in \mathbb{R}^n$, where each element $\boldsymbol{g}_{d,t}$ corresponds to the gate value for the $t$-th text token. To mitigate the influence of the cross-attention branch during early training stages, a static, learnable warm-up factor $g_s$ is introduced, initialized to zero.

The final fused representation $\boldsymbol{X}_t$ is obtained by modulating and merging the cross-attention output $\boldsymbol{X}'$ into the original text embeddings $\boldsymbol{X}_t$ as follows:

$$\boldsymbol{X}_t = \boldsymbol{X}_t + \boldsymbol{X}' \odot \boldsymbol{g}_d \cdot g_s \tag{12}$$

where $\odot$ denotes element-wise multiplication broadcasted along the token dimension. Importantly, the cross-attention module updates only $\boldsymbol{X}_t$, the text-only component of the hidden states, while the scene tokens remain fixed throughout this process.

## 4.3 WEIGHT INITIALIZATION.

As detailed in Section 3, the formulations of RWKV and linear attention can be approximated by equation(7) and equation(4), respectively. These two operations exhibit several structural similarities. To facilitate analysis, we rewrite equation(7) and equation(4) into a unified form as follows:

$$\boldsymbol{S}_t = \boldsymbol{S}_{t-1} + \boldsymbol{k}_t^\top \boldsymbol{v}_t, \qquad\qquad \boldsymbol{y}_t = \boldsymbol{r}_t \boldsymbol{S}_t / \mathbb{1}. \tag{13}$$

$$\boldsymbol{S}_i = \boldsymbol{S}_{i-1} + \boldsymbol{K}_i^\top \boldsymbol{V}_i, \qquad\qquad \boldsymbol{y}_i = \boldsymbol{Q}_i \boldsymbol{S}_i / \boldsymbol{Q}_i \boldsymbol{Z}_i. \tag{14}$$

It is evident that equation(13) and equation(14) are highly similar. Specifically: $k \sim K$, $v \sim V$, $r \sim Q$. To maximize parameter reuse and minimize training cost, we initialize the RWKV projection matrices $W_r$, $W_k$, and $W_v$ using the corresponding pre-trained self-attention weights $W_Q$, $W_K$, and $W_V$, respectively. Similarly, the projection weights of the cross-attention module ($W_Q$, $W_K$, $W_V$) are also initialized from the same set of pre-trained self-attention parameters. (Wang et al., 2024a) suggests that the majority of knowledge in Transformer-based models is encoded within the MLP layers. Motivated by this finding, we retain both the architectural structure and pre-trained weights of the MLP layers to preserve this transferred knowledge and avoid unnecessary retraining.

### 4.4 PROGRESSIVE DISTILLATION

To more effectively inherit and leverage the powerful representational capacity of the pre-trained model, we adopt a progressive knowledge distillation(Hinton et al., 2015) strategy. The distillation process is partitioned into three interrelated yet distinct stages: (1) RWKV Distillation, (2) Decoder Layer Distillation, and (3) LLM Distillation. This approach gradually increases both the duration of training videos and the number of trainable parameters, guiding the student model to transfer knowledge from structural components to holistic language modeling capabilities, thereby enhancing its generalization and task adaptability.

**Stage 1: RWKV Distillation.** In this stage, the primary objective is to align the newly introduced RWKV modules and cross-attention components, which lack pre-trained weights, with their corresponding self-attention counterparts in the original architecture. During training, all model parameters except those of the RWKV and cross-attention modules are frozen, including remaining self-attention layers, MLP layers within replaced decoder blocks, and the visual encoder. For this stage, we sample image-caption pairs to train the model's image understanding capability. Additionally, we sample short video clips to enhance its short video understanding ability. The training objective is to minimize the Kullback-Leibler divergence (KLD) between the output logits of the student (hybrid) model and the teacher (original pre-trained) model:

$$\mathcal{L}_{\text{kd}}(\pi_S; \pi_T) = -\mathbb{E}_{(x,y_k) \sim \pi_T} \left[ \log \frac{\pi_T(y_k \mid y_{<k}, x)}{\pi_S(y_k \mid y_{<k}, x)} \right] \tag{15}$$

where $\pi_T(y_k \mid y_{<k}, x)$ and $\pi_S(y_k \mid y_{<k}, x)$ denote the probability of the predicted tokens for teacher MLLM and student MLLM.

**Stage 2: Decoder Layer Distillation**. In this stage, the goal is to enable the newly introduced hybrid decoder layers to better mimic the output distribution of their original counterparts. All model parameters are frozen except those of the hybrid decoder layers. The training objective remains consistent with Stage 1. The training data comprises newly sampled short videos and medium-length videos.

**Stage 3: LLM Distillation.** During this stage, we unfreeze all parameters in the LLM component, while keeping the visual encoder frozen to maintain stable visual representations. The training dataset includes a long video dataset based on short and medium videos. To fully leverage both the knowledge embedded in the pre-trained model and the supervisory signal from downstream data, we employ a dual-objective strategy that combines the Kullback-Leibler divergence (KLD) loss with the standard next-token prediction objective. Specifically, the KLD loss aligns the student's output distribution with that of the teacher to preserve pre-trained knowledge, while the next-token prediction loss optimizes task-specific performance using ground-truth supervision. The total loss is formulated as:

$$\mathcal{L}_{\text{LLM}}(\pi_S; \pi_T) = -\alpha \mathbb{E}_{(x,y_k) \sim \mathcal{D}} \left[ \log \pi_S(y_k \mid y_{<k}, x) \right] - \beta \mathbb{E}_{(x,y_k) \sim \pi_T} \left[ \log \frac{\pi_T(y_k \mid y_{<k}, x)}{\pi_S(y_k \mid y_{<k}, x)} \right] \tag{16}$$

where, $\mathcal{D}$ denotes the training data distribution, $\alpha$ and $\beta$ are balancing coefficients, balancing task-specific learning and knowledge preservation.

## 5 EXPERIMENTS

### 5.1 EXPERIMENTAL SETUP

**Target Models.** We adopt Qwen2.5-VL-7B-Instruct (Bai et al., 2025) as the base model for our experiments. We replace selected decoder layers with our proposed hybrid RWKV decoder layers.

Specifically, we first evaluate a configuration where 25% of the decoder layers are replaced. Building upon this, we further apply a layer-wise replacement strategy to substitute an additional 25% of layers, resulting in a total of 50% replaced layers for comparative analysis. We set the maximum boundary count $K = 15$ and the similarity threshold $\tau = 0.60$.

**Training Recipe.** We employ a three-stage distillation strategy to enable the hybrid model to progressively acquire knowledge from the original model. In each stage, we expand the training data to include longer video sequences and adjust the training configuration accordingly. All experiments are conducted on a machine equipped with 8 NVIDIA A800 80G GPUs. Detailed training schedules and hyperparameters are specified in Appendix A.1.

**Evaluation Benchmarks.** We evaluate our method on several mainstream video understanding benchmarks. To assess the model's needle-in-a-haystack capability, we use VNBench(Zhao et al., 2024b). For long video understanding, we adopt MLVU (Zhou et al., 2025), Video-MME (Fu et al., 2025), LongVideoBench (Wu et al., 2024), and LVBench(Wang et al., 2024c).

## 5.2 MAIN EVALUATION RESULTS

Table 1: Results on VNBench for needle-in-a-haystack capability. The best result in each column is **highlighted in bold**. Models marked with an asterisk ($^*$) are evaluated using our own evaluation script.

| Model | Size | Retrieval | | | Ordering | | | Counting | | | Avg |
|---|---|---|---|---|---|---|---|---|---|---|---|
| | | E | I-1 | I-2 | E | I-1 | I-2 | E-1 | E-2 | I | |
| **Proprietary Models** | | | | | | | | | | | |
| Gemini 1.5 Pro (Team et al., 2024) | - | **100.0** | 96.0 | 76.0 | **90.7** | **95.3** | 32.7 | **60.7** | 7.3 | **42.0** | **66.7** |
| GPT-4o (OpenAI, 2024) | - | **100.0** | 98.0 | 87.3 | 88.4 | 86.6 | **45.2** | 36.8 | 0.0 | 36.1 | 64.4 |
| GPT-4V (Achiam et al., 2023) | - | **100.0** | 99.3 | 82.0 | 42.6 | 22.8 | 23.0 | 37.6 | 0.0 | 32.4 | 48.9 |
| **Open-source MLLMs** | | | | | | | | | | | |
| VideoChatGPT (Maaz et al., 2023) | 7B | 4.7 | 4.7 | 0.7 | 2.7 | 11.3 | 0.0 | 2.0 | 4.0 | 6.7 | 4.1 |
| Video-LLaMA2 (Zhang et al., 2023) | 7B | 1.2 | 26.0 | 6.0 | 0.0 | 0.0 | 0.0 | 2.0 | 4.7 | 0.7 | 4.5 |
| Video-LLaVA-7B (Lin et al., 2023) | 7B | 26.0 | 28.0 | 17.3 | 0.7 | 0.7 | 2.0 | 16.7 | 0.7 | 20.0 | 12.4 |
| VideoChat2 (Li et al., 2023b) | 7B | 43.4 | 40.0 | 14.6 | 0.0 | 0.0 | 1.3 | 3.3 | 0.7 | 8.0 | 12.4 |
| LLaVA-NeXT-Video-7B (Liu et al., 2024b) | 7B | 56.7 | 56.7 | 19.3 | 0.7 | 0.0 | 0.7 | 6.7 | 14.6 | 25.3 | 20.1 |
| ST-LLM (Liu et al., 2024d) | 7B | 58.0 | 64.7 | 31.3 | 0.0 | 0.0 | 0.0 | 21.3 | 1.3 | 27.3 | 22.7 |
| LLaVA-OneVision-7B (Li et al., 2024a) | 7B | 88.7 | **87.3** | 55.3 | 70.0 | 50.0 | 37.3 | 41.3 | 8.7 | 27.3 | 51.8 |
| Qwen2.5-VL-7B$^*$ (Bai et al., 2025) | 7B | **100.0** | 78.8 | **97.8** | 91.6 | 85.2 | 85.2 | 67.4 | 23.1 | 30.7 | 73.4 |
| **Hybrid-architecture MLLMs** | | | | | | | | | | | |
| LongLLaVA(Wang et al., 2024d) | 9B | 98.3 | 57.2 | 96.3 | 24.2 | 57.2 | 24.3 | 24.5 | 21.0 | 26.0 | 44.4 |
| **Qwen-RWKV-VL(25%)(ours)** | 8B | **100.0** | 70.2 | **98.8** | 91.8 | 89.3 | 79.1 | 71.4 | 20.4 | **36.6** | **74.0** |
| **Qwen-RWKV-VL(50%)(ours)** | 9B | 99.8 | 60.8 | 98.7 | 41.0 | 57.6 | 58.1 | 55.9 | **22.7** | 34.9 | 58.6 |

**Needle-in-a-Haystack Performance.** As shown in Table 1, our hybrid model with 25% of decoder layers replaced by RWKV modules surpasses the original model by a small margin on the needle-in-a-haystack task. Moreover, by leveraging pre-trained weights from the original model, our approach achieves this superior performance using significantly less training data, substantially outperforming models trained from scratch such as LongLLaVA.

**Video Understanding Performance.** We compare our model against other state-of-the-art video understanding models in Table 2. For our hybrid architecture, we evaluate two configurations: replacing 25% and 50% of decoder layers with RWKV modules, respectively. Notably, both VAMBA (Ren et al., 2025) and Slow-fast MLLM (Shi et al., 2025) exhibit substantial performance degradation compared to their original counterparts, despite being trained on over 6M and 3,467K samples, respectively. In contrast, our 25% replacement model achieves competitive or superior results relative to the baseline across most benchmarks, with particularly pronounced improvements on LVBench and VNBench, while requiring only 1,448K training samples.

## 5.3 ABLATION STUDY

**Replacement Layer Position.** To study the impact of RWKV layer placement, we replace layers every 7th position in three regions: *early* (0, 7, 14, 21), *middle* (3, 10, 17, 24), and *late* (6,

Table 2: Results of video understanding evaluation. The best result in each column is **highlighted in bold**. Models marked with an asterisk (*) are evaluated using our own evaluation script.

| Model | Size | VideoMME$_{\text{w/o subs}}$ | | | | MLVU m-avg | LongVideoBench val | LVBench test |
| | | Short | Medium | Long | Avg | | | |
|---|---|---|---|---|---|---|---|---|
| **Proprietary Models** | | | | | | | | |
| Gemini 1.5 Pro (Team et al., 2024) | - | **81.7** | **74.3** | **67.4** | **75.0** | 62.9 | 64.0 | 33.1 |
| GPT-4o (OpenAI, 2024) | - | 80.0 | 70.3 | 65.3 | 71.9 | **64.6** | **66.7** | **48.9** |
| GPT-4V (Achiam et al., 2023) | - | 70.5 | 55.8 | 53.5 | 59.9 | 49.2 | 59.1 | 48.7 |
| **Open-source MLLMs** | | | | | | | | |
| mPLUG-Owl3 (Ye et al., 2024) | 7B | 70.0 | 57.7 | 50.1 | 59.3 | - | 52.1 | 43.5 |
| Video-LLaMA2 (Zhang et al., 2023) | 7B | 55.9 | 45.4 | 42.1 | 47.8 | 32.7 | - | - |
| Video-LLaVA-7B (Lin et al., 2023) | 7B | 45.3 | 38.0 | 36.2 | 39.9 | 47.3 | 39.1 | - |
| VideoChat2 (Li et al., 2023b) | 7B | 48.3 | 37.0 | 33.2 | 39.5 | 47.9 | 36.0 | - |
| Kangaroo (Liu et al., 2024c) | 8B | 66.1 | 55.3 | 46.6 | 56.0 | 61.0 | 56.0 | 38.3 |
| LLaVA-OneVision-7B (Li et al., 2024a) | 7B | - | - | - | 58.2 | 64.7 | **56.5** | 38.7 |
| Qwen2.5-VL-7B* (Bai et al., 2025) | 7B | **76.1** | **61.7** | **53.6** | **63.8** | 70.4 | 49.5 | **45.3** |
| **Hybrid-architecture MLLMs** | | | | | | | | |
| LongLLaVA(Wang et al., 2024d) | 9B | 52.4 | 42.2 | 36.4 | 43.7 | 53.3 | 42.1 | 31.2 |
| VAMBA(Ren et al., 2025) | 10B | - | - | - | 57.8 | 65.9 | 55.9 | 42.1 |
| Slow-fast MLLM(Shi et al., 2025) | 8B | - | - | - | 60.3 | **68.1** | **58.0** | - |
| **Qwen-RWKV-VL(25%)(ours)** | 8B | **74.4** | **56.9** | **52.4** | **61.3** | 68.0 | 47.8 | **46.8** |
| **Qwen-RWKV-VL(50%)(ours)** | 9B | 62.9 | 47.2 | 40.6 | 50.2 | 61.2 | 39.9 | 43.2 |

13, 20, 27), keeping other layers as Transformer blocks. All variants are trained under identical distillation settings and evaluated on Video-MME. As shown in Table 3 left, middle-layer and late-layer replacements achieve comparable performance, while early-layer replacement causes significant degradation. We attribute this to disruption of foundational language modeling in early layers, which harms information propagation and representation learning. This also clarifies why 50% layer replacement causes such a dramatic decrease: excessive substitutions, particularly in early network segments, fundamentally weaken the model's representational power. All subsequent experiments are conducted on the setting where the late-layer is replaced.

Table 3: Ablation studies on (left) layer replacement positions and (right) loss coefficients $\alpha, \beta$. Performance reported on Video-MME. Best results per column are **bolded**.

| Position | Short | Medium | Long | Avg | $(\alpha, \beta)$ | Short | Medium | Long | Avg |
|---|---|---|---|---|---|---|---|---|---|
| early | 16.0 | 16.1 | 16.2 | 16.1 | (1.0, 0.0) | 71.3 | 55.8 | 48.3 | 58.5 |
| middle | 68.9 | **53.7** | **48.9** | **57.1** | (0.5, 0.5) | 71.2 | 55.4 | 50.2 | 59.0 |
| late | **70.7** | 53.1 | 47.7 | **57.1** | (0.0, 1.0) | **74.4** | **56.9** | **52.4** | **61.3** |

**Loss Balancing Coefficients $\alpha$ and $\beta$.** To analyze the sensitivity of our hybrid distillation objective to the weighting of its components, we conduct an ablation study over different values of $\alpha$ and $\beta$ in the combined loss $\mathcal{L}_{\text{KD}} = \alpha \cdot \mathcal{L}_{\text{CE}} + \beta \cdot \mathcal{L}_{\text{KL}}$. We evaluate configurations including $(\alpha, \beta) = (1.0, 0.0)$, $(0.5, 0.5)$, $(0.0, 1.0)$. Results on Video-MME. As shown in Table 3 right, the $(0.0, 1.0)$ configuration, which uses only distillation loss without SFT supervision, achieves the best performance. This stems from limited SFT data quality: adding SFT loss introduces noise that disrupts knowledge alignment with the teacher. In contrast, distillation loss alone enables the student to better match the teacher's output distribution, yielding more stable and consistent transfer.

**Gate.** We compare two gating mechanisms on Video-MME: a static gate that applies a single learnable scalar to all text tokens, and our proposed dynamic gate that predicts a distinct gating value for each token based on its embedding. As shown in Table 4, comparing rows 2 and 4, the dynamic gate consistently outperforms the static variant on Video-MME benchmark. This demonstrates that token-adaptive modulation is critical for fine-grained fusion of visual and textual features.

**Weight Initialization.** We compare two RWKV initialization strategies on Video-MME: Random (random init) and Transfer (weights mapped from self-attention). As shown in Table 4 (row 1 vs 2), Transfer improves average performance from 60.6% to 61.3%, demonstrating that mapped pretrained weights provide stronger inductive bias than random initialization, better preserving knowledge during distillation.

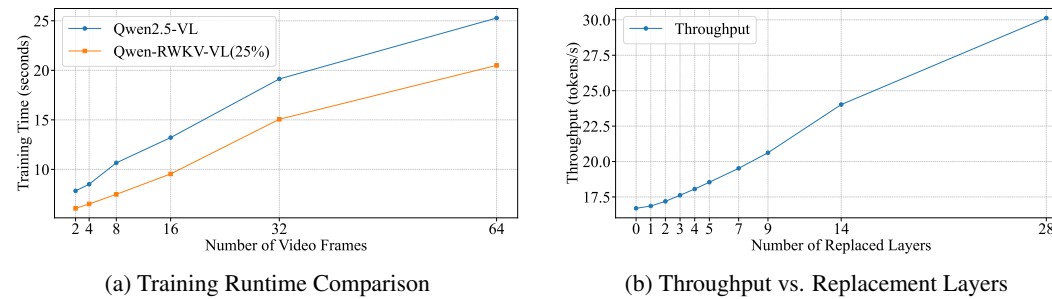

(a) Training Runtime Comparison    (b) Throughput vs. Replacement Layers

Figure 2: Comparison of training speed between Qwen2.5VL and RWKV hybrid architecture with different input frame numbers, and throughput comparison between different numbers of replacement layers.

**Cross-attention.** We compare two variants on Video-MME: without cross-attention (RWKV-only) and with cross-attention for global visual context. As shown in Table 4 (row 2 vs 3), cross-attention boosts average performance from 58.6% to 61.3%, demonstrating that while RWKV captures local dependencies, it lacks long-range visual grounding. Cross-attention provides global anchors that mitigate history decay—critical for long videos.

Table 4: Comparisons of different gate designs, weight initializations and cross-attention Integration. Performance reported on Video-MME. Best results per column are **bolded**.

| Gate | Init | Cross_attn | Short | Medium | Long | Avg |
|---|---|---|---|---|---|---|
| Dynamic | Random | ✓ | 72.2 | **58.3** | 51.2 | 60.6 |
| Dynamic | Transfer | ✓ | **74.4** | 56.9 | **52.4** | **61.3** |
| Dynamic | Transfer | ✗ | 71.3 | 56.6 | 47.8 | 58.6 |
| Static | Transfer | ✓ | 72.5 | 57.3 | 51.1 | 60.3 |

## 5.4 EFFICIENCY ANALYSIS

To quantify the runtime efficiency gains of our hybrid model over the baseline Transformer-based MLLM (Qwen2.5-VL-Instruct-7B), we conduct comprehensive training and inference benchmarks under controlled conditions. For training, we ensure a consistent environment by using 8 NVIDIA A800 80GB GPUs for both models, with a per-GPU batch size of 64. We apply standard optimization techniques, including FlashAttention-2 (Dao, 2023), DeepSpeed ZeRO-3 (Rajbhandari et al., 2020), and gradient checkpointing (Chen et al., 2016). We measure the wall-clock time per training step across varying input frame counts. As shown in Figure 2a, our model achieves nearly 25% faster training at 64 input frames.

For inference, we focus on throughput, defined as the number of tokens generated per second. We fix the input text length to 10K tokens and measure the time to generate the first token ($t_1$) and the 1000th token ($t_{1000}$). Throughput is computed as $(1000 - 1)/(t_{1000} - t_1)$. To better simulate real-world deployment, both models are evaluated on a single NVIDIA A800 80GB GPU with FlashAttention-2 enabled. As shown in Figure 2b, replacing all Transformer layers with RWKV modules nearly doubles inference throughput, while replacing only 25% of layers (7 layers) still yields a 20% speedup.

## 6 CONCLUSION

We presented a hybrid RWKV-Transformer architecture for efficient multimodal video understanding. By integrating RWKV modules to replace selected Transformer layers and incorporating cross-attention mechanisms for global context modeling, our approach significantly improves inference throughput while maintaining competitive performance across multiple video understanding benchmarks. Extensive evaluations on datasets such as VNBench and LVBench demonstrate our model's superiority over existing efficient video LMMs, particularly in handling hour-long video inputs.

ETHICS STATEMENT

The authors have read and adhere to the ICLR Code of Ethics. This work does not involve human subjects, identifiable private data, or harmful applications. All datasets used are publicly available and were used in accordance with their original licenses and intended purposes. No external sponsorship or conflict of interest influenced the design or conclusions of this work.

REPRODUCIBILITY STATEMENT

All code and source files are provided in the supplementary material and will be publicly released. Additional implementation details can be found in Appendix A. Besides additional experimental results and qualitative evaluations are provided in the appendix. Specifically, Appendix B includes ablation studies on the effectiveness of progressive training stages and scene segmentation hyperparameters, while Appendix C presents qualitative results on object recognition, visual reasoning, needle in a haystack, and counting problems, along with examples.

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

# A IMPLEMENTATION DETAILS

## A.1 TRAINING STRATEGY AND HYPERPARAMETERS

We adopt a three-stage progressive training strategy to stabilize knowledge transfer and enable hierarchical alignment. In Stage 1, we freeze all modules except the RWKV and cross-attention components (which lack pre-trained weights), and train the model on image and short-video datasets to establish basic visual recognition capability. In Stage 2, we unfreeze all inserted RWKV decoder layers while keeping other parameters frozen, and extend the training data to include medium-length videos, enabling layer-wise alignment between the hybrid and original architectures. In the final stage, we freeze only the ViT encoder and multimodal projector, unfreeze the entire LLM, and incorporate long videos into the training mix to achieve full model-level alignment. All training experiments are conducted using the ms-swift framework (Zhao et al., 2024a). Detailed training hyperparameters are provided in Table 5.

Table 5: Training hyperparameters for each stage.

| Hyperparameter | Stage 1 | Stage 2 | Stage 3 |
|---|---|---|---|
| Batch size | 64 | 64 | 64 |
| Learning rate | $1 \times 10^{-4}$ | $1 \times 10^{-5}$ | $1 \times 10^{-6}$ |
| Weight decay | 0.1 | 0.1 | 0.1 |
| Warmup ratio | 0.03 | 0.03 | 0.03 |
| Video max pixels | 50176 | 50176 | 50176 |
| Fps max frames | 64 | 64 | 64 |
| Training epochs | 1 | 1 | 1 |
| Optimizer | AdamW | AdamW | AdamW |
| Optimizer hyperparameter | $\beta_1 = 0.9, \beta_2 = 0.95$ | $\beta_1 = 0.9, \beta_2 = 0.95$ | $\beta_1 = 0.9, \beta_2 = 0.95$ |
| Learning rate schedule | Cosine | Cosine | Cosine |
| Max sequence length | 8192 | 8192 | 8192 |
| Number of samples | 542K | 656K | 250K |
| Trainable parameters | RWKV + Cross-attention | Hybrid decoder layers | Full LLM |
| Loss function | KL divergence | KL divergence | $\alpha$CE + $\beta$KL |
| $\alpha$ / $\beta$ weights | - | - | 0.0 / 1.0 |
| Mixed precision | BF16 | BF16 | BF16 |
| Model parallelism | Zero2 offload | Zero3 | Zero3 offload |

## A.2 TRAINING DATASETS

Our training methodology employs a progressive data strategy across three distinct stages. The initial stage utilizes 542k samples, comprising 342k image-caption pairs sampled from ShareGPT4V-PT(Chen et al., 2024a) for visual understanding enhancement and 200k samples from 0-30s video clips in LLaVA-Video-178k(Zhang et al., 2024c) for short video comprehension training, focusing on distilling RWKV and cross-attention modules. The second stage expands the dataset to 656k samples by incorporating one-third of the 0-30s video clips and half of the 30-60s video clips from LLaVA-Video-178k, targeting the distillation of hybrid decoder layers. The final stage further diversifies the training corpus to 250k samples by adding 1-3min video segments, enabling end-to-end fine-tuning of the LLM component while preserving visual encoder parameters frozen. Throughout this stage, we employ a hybrid objective function combining Kullback-Leibler divergence loss with standard next-token prediction cross-entropy loss to optimally balance knowledge preservation from the pre-trained model and adaptation to downstream task requirements.

## A.3 EVALUATION BENCHMARKS

**LVBench** (Wang et al., 2024c) is a benchmark designed to evaluate the capability of video LMMs in understanding ultra-long videos. It contains 1,549 question-answer pairs, with an average video duration of 4,101 seconds. Evaluation focuses on six core dimensions: (1) *temporal grounding*, which assesses the model's ability to identify specific moments in videos; (2) *video summarization*, evaluating the capacity to condense key information; (3) *video reasoning*, testing logical inference

over video content; (4) *entity recognition*, identifying persons, objects, or locations; (5) *event understanding*, capturing the sequence and significance of events; and (6) *key information retrieval*, ensuring the model can extract critical details. The full test set is used for evaluation.

**VNBench** (Zhao et al., 2024b) is a synthetic video benchmark constructed using the VideoNIAH framework, designed to evaluate fine-grained video understanding capabilities under controlled conditions. It decouples visual content from query-response pairs by inserting synthetic "needles" (unrelated visual elements) into source videos, enabling targeted evaluation of specific skills. The benchmark includes three core tasks: (1) *Retrieval* — locating specific moments; (2) *Ordering* — reasoning about event chronology; and (3) *Counting* — tracking object occurrences over time. These tasks assess temporal perception, chronological reasoning, and spatio-temporal coherence, respectively. By automating query generation and supporting variable video lengths, VNBench offers a scalable, skill-isolating evaluation platform ideal for iterative model development.

**Video-MME** (Fu et al., 2025) is a benchmark specifically designed to evaluate the ability of LMMs to analyze video content. It comprises a dataset of 900 videos and 2,700 questions, covering six distinct visual domains. Questions are categorized by video length into short, medium, and long segments, with median durations of 26 seconds, 164.7 seconds, and 890.7 seconds, respectively. The benchmark supports two evaluation settings: (1) *w/subtitle*, where both subtitles and questions are provided as text input, and (2) *w/o subtitle*, which relies solely on raw video input alongside the question. Our study primarily focuses on the *w/o subtitle* setting, as it encourages models to leverage video-based visual cues rather than textual prompts from subtitles, thereby providing a more authentic evaluation of long-video understanding capability.

**MLVU** (Zhou et al., 2025) is a benchmark designed to evaluate long-video understanding across diverse tasks and video genres. It includes two types of questions: multiple-choice and free-form generation. The evaluation framework measures LMM performance along three key dimensions: (1) *holistic video understanding*, which requires reasoning over the global context of the entire video; (2) *single-detail video understanding*, focusing on identifying critical moments or short segments; and (3) *multi-detail video understanding*, which involves establishing connections among multiple short segments distributed throughout the video.

**LongVideoBench** (Wu et al., 2024) is a question-answering benchmark designed for interleaved long video-text inputs. It contains 3,763 videos and 6,678 human-annotated multiple-choice questions, covering 17 fine-grained categories. The benchmark supports two input formats: (1) *mat format*, which processes all video tokens first followed by the question text; and (2) *interleaved video-text format*, which inserts subtitles between video frames to simulate natural multimodal streaming. In our evaluation, we adopt the standard mat format for all baseline models and our VAMBA. Reported results are based on the validation split to ensure fair and reproducible comparison.

# B  ADDITIONAL EXPERIMENTS

## B.1  EFFECTIVENESS OF PROGRESSIVE TRAINING STAGES

We conduct an ablation study to evaluate the contribution of each training stage in our progressive distillation strategy.

Table 6: Ablation study on the effectiveness of progressive training stages. Performance reported on Video-MME. Each strategy builds upon the previous one except only stage2. Best results per column are **bolded**.

| Qwen-RWKV-VL(25%) | Short | Medium | Long | Avg |
|---|---|---|---|---|
| Stage 1 (RWKV + Cross-Attn) | 67.7 | 51.8 | 43.4 | 54.3 |
| + Stage 2 (Hybrid Decoder Layers) | 71.1 | 56.1 | 48.1 | 58.4 |
| + Stage 3 (Full LLM) | **74.4** | **56.9** | **52.4** | **61.3** |
| Only Stage 2 | 63.8 | 46.4 | 38.9 | 49.7 |

Table 6 demonstrates the necessity of our staged training strategy. Starting with Stage 1 (training only RWKV and cross-attention modules) establishes a performance baseline (54.3% Avg). Adding Stage 2 (unfreezing hybrid decoder layers) improves Avg to 58.4%, and completing Stage 3 (full LLM fine-tuning) further boosts performance to 61.3%, confirming that gradual exposure to longer videos and progressive parameter unfreezing are essential for stable knowledge transfer.

Critically, the "Only Stage 2" configuration where hybrid decoder layers are trained from scratch without first stabilizing the non-pretrained components in Stage 1 performs significantly worse (49.7% Avg). This degradation occurs because directly optimizing decoder layers without prior alignment of RWKV and cross-attention modules disrupts the pretrained MLP weights, effectively "polluting" the inherited linguistic and visual priors.

## B.2  SCENE SEGMENTATION HYPERPARAMETERS

We further investigate the impact of varying $K$ and similarity threshold $\tau$ on model performance.

Table 7: Ablation study on scene segmentation hyperparameters: similarity threshold $\tau$ and maximum number of scenes ($K + 1$), where $K$ is the maximum number of boundary positions selected. Performance reported on VNBench. Best results per column are **bolded**.

| Threshold $\tau$ | Max Scenes ($K + 1$) | Retrieval | Ordering | Counting | Avg |
|---|---|---|---|---|---|
| 0.85 | 8 | **90.3** | 84.8 | 40.6 | 71.9 |
| 0.85 | 16 | 88.4 | 86.2 | 41.8 | 72.2 |
| 0.60 | 16 | 89.9 | **86.6** | **43.5** | **74.0** |
| 0.50 | 16 | 89.2 | **86.6** | 42.3 | 73.0 |

As shown in Table 7, the configuration with $\tau = 0.60$ and $K + 1 = 16$ achieves the highest average score (74.0%), outperforming both stricter thresholds like $\tau = 0.85$ and looser ones like $\tau = 0.50$. Under a fixed threshold ($\tau = 0.85$), increasing the maximum scene count from 8 to 16 slightly improves performance (71.9% → 72.2%), suggesting that allowing more scene segments helps preserve finer visual transitions. In contrast, $\tau = 0.60$ with $K + 1 = 16$ achieves the best balance, as it permits more semantically meaningful boundaries while avoiding excessive fragmentation. Lowering $\tau$ further to 0.50 (still with 16 scenes) reduces performance (73.0%), indicating that overly permissive thresholds introduce noisy or redundant segments that harm global representation quality.

# C QUALITATIVE EVALUATION

## C.1 PROMPTS FOR EVALUATION

We specify distinct prompts for each evaluation benchmark. For multiple-choice benchmarks, we adopt a unified prompt format as follows:

---

**Evaluation prompt for multi-choice question answering benchmarks.**
<Video>
Select the best answer to the following multiple-choice question based on the video and the subtitles. Respond with only the letter (A, B, C, or D) of the correct option.
<Question>
A. <Option 1>
B. <Option 2>
C. <Option 3>
D. <Option 4>
Other options · · ·
The best answer is:

---

Figure 3: Prompt for multiple-choice benchmarks

For the open-ended benchmarks, we use the following format as below:

---

**Evaluation prompt for open-ended benchmarks.**
<Video>
<Question>
Answer the question using a single word or phrase.

---

Figure 4: Prompt for open-ended benchmarks

## C.2 QUALITATIVE RESULTS

In this section, we present additional qualitative results of our hybrid model, including tasks such as object recognition, visual reasoning, needle in a haystack, and counting problems. Results are visualized in Figure 5.

————————————————Object Recognition————————————————

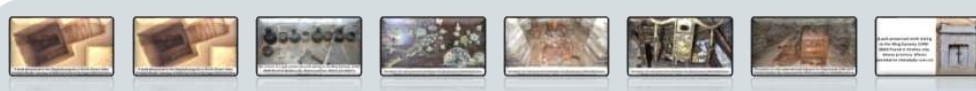

**Question:** Which of the following features/items is not discussed in the video in relation to the tomb?
**Options:** 'A. Inkstone.', 'B. Niche.', 'C. Jade.', 'D. Sacrificial table.'
**Model prediction:** C

————————————————Visual Reasoning————————————————

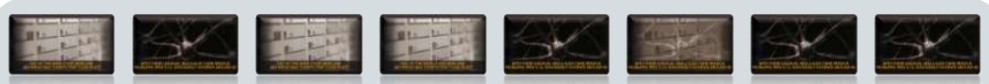

**Question:** On the screen, there are many advanced math problems on a piece of white paper. Among the problems, there is also a complete brain diagram. When the subtitle mentions 'an extraordinary brain that lacked the,' what color appears on the right side of the brain diagram?
**Model prediction:** The right side of the brain is blue with stains.

————————————————Needle in a Haystack————————————————

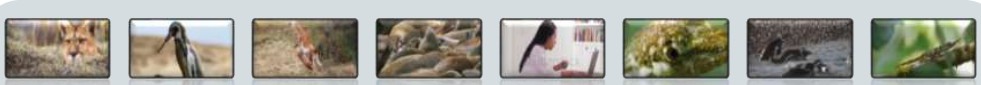

**Question:** What is the woman's appearance?
**Options:** 'A. She is an elderly woman.', 'B. She is a young girl.', 'C. She is a middle-aged woman.', 'D. She is a beautiful woman.'
**Model prediction:** D

————————————————Counting Problem————————————————

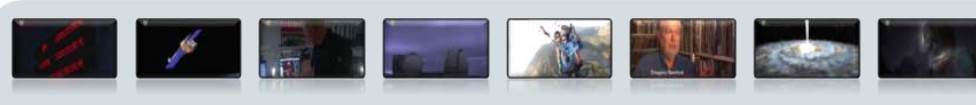

**Question:** Throughout this video, what is the total count of occurrences for the scene featuring the 'paragliding' action?
**Model prediction:** 1.

Figure 5: Visualization results of our model in some video understanding tasks.

# D  THE USE OF LARGE LANGUAGE MODELS (LLMs)

We disclose that we used Qwen3-Max-Preview(Yang et al., 2025) to assist in polishing the language and improving the clarity of this paper. The model was used for grammar correction, sentence restructuring, and enhancing overall readability. All technical content, experimental design, results, and conclusions were authored and verified solely by the human authors. The LLM did not contribute to the generation of ideas, methods, or data analysis.

