# OpenReview forum: "Bridging Transformers and RWKV: Towards Efficient Multimodal Video Understanding"
_ICLR.cc/2026/Conference — Submitted to ICLR 2026_

### Official Review · Reviewer_FGe7 · 2025-10-27

**Soundness:** 3
**Presentation:** 3
**Contribution:** 2
**Rating:** 2
**Confidence:** 4

**Summary:**

The paper proposes a hybrid RWKV-Transformer architecture to mitigate the quadratic computational complexity of processing long videos in MLLMs. The core idea is to replace a portion of the Transformer's self-attention layers with linear-complexity RWKV modules. To make this feasible, the authors introduce three main contributions: (1) a weight initialization strategy that remaps pre-trained attention weights ($W_Q, W_K, W_V$) to the new RWKV modules ($W_r, W_k, W_v$); (2) a parallel cross-attention mechanism that uses "scene tokens" to provide global context and mitigate RWKV's history decay; and (3) a three-stage progressive distillation process to train the hybrid model. The authors claim their model, when replacing 25% of layers, can match or exceed the baseline's performance while improving throughput by 20%.

**Strengths:**

1. The paper tackles the critical and well-known challenge of $O(N^2)$ complexity in long-context video modeling, which is a significant bottleneck for MLLMs.

2. The idea of re-using the pre-trained $W_Q, W_K, W_V$ weights to initialize the RWKV modules is a smart approach to leverage the knowledge from the original model and reduce the training cost and instability of introducing new, randomly-initialized architectures.

3. The paper provides informative ablation studies. For example, the study on replacement layer position (Table 3) clearly shows that replacing late layers is effective while replacing early layers is detrimental. Furthermore, the ablations in Table 4 effectively demonstrate the necessity of both the weight transfer and the cross-attention module.

**Weaknesses:**

1. The central claim that the 25% model "match[es] its performance on multiple video understanding benchmarks"  is an overstatement. While it does show marginal gains on VNBench (+0.6%) and LVBench (+1.5%), it shows clear performance degradation on other key benchmarks compared to the Qwen2.5-VL-7B baseline. On Video-MME, the hybrid model scores 61.3% (Avg), which is 2.5 points lower than the baseline's 63.8%. On MLVU, the hybrid model scores 68.0% (m-avg), which is 2.4 points lower than the baseline's 70.4%. This is not "matching" performance; it is a clear trade-off (sacrificing performance on some benchmarks for gains on others) that is not adequately acknowledged in the abstract or introduction.

2. The efficiency gains are presented in a potentially misleading manner. The abstract and introduction prominently claim a throughput increase of "up to nearly 2x". However, this 2x gain is only achieved when all Transformer layers are replaced (as shown in Figure 2b). The performance of this 100%-replacement model is never reported, and the catastrophic failure of the 50%-replacement model  strongly implies the 100% model's performance would be unusable. The actual model being advocated for (the 25% variant) achieves only a 20% throughput gain. This is a modest improvement, not a 2x one.

3. The proposed method appears to be extremely brittle. The ablations reveal that the approach only works in a very specific "sweet spot":
- It fails when replacing early layers.
- It fails when replacing 50% of layers, causing a "dramatic decrease" in performance. This suggests the method is not a robust, generalizable hybridization strategy. Instead, it's a highly-tuned workaround that breaks if a user attempts to push for more efficiency by replacing more layers. This severely limits the method's primary utility.

4. The 20% inference speedup (which comes with a performance trade-off, not a clear win) is achieved at the cost of a significant increase in training and architectural complexity.

**Questions:**

Please see Weaknesses. While the paper presents an interesting direction for hybridizing models, the execution is not convincing. The performance claims are overstated (the model underperforms the baseline on several key benchmarks), the efficiency gains of the usable model are modest (20%, not 2x), and the method itself appears brittle, complex, and difficult to scale. Therefore, the paper in its current form falls below the acceptance threshold.

---

> ### Author Response · Authors · 2025-11-24
> **Rebuttal to Reviewer FGe7**
>
> We appreciate the reviewer's insightful comments and we address the concerns below:
>
> - **[W1] The central claim that the 25% model "match[es] its performance on multiple video understanding benchmarks" is an overstatement. **:
>
>   - Since the video-training data used for Qwen2.5-VL is not publicly available, our experiments are necessarily conducted on the existing open-source datasets. Even under these constraints, our hybrid model—trained with **significantly fewer samples**—achieves performance that **surpasses the original model on several benchmarks** and shows only **minor degradation** on others. We believe this level of performance should reasonably be considered *comparable* to the full-attention baseline.
>
>     Moreover, for a fair comparison among **hybrid architectures**, it is more appropriate to evaluate our approach against models such as **VAMBA** and **Slow-Fast MLLM**. Both of these methods *retain* the original attention layers and parameters, while simply adding Mamba or cross-attention modules on top. In contrast, our method performs a **much more challenging architectural replacement**—substituting self-attention with RWKV—yet still achieves **stronger video-understanding performance with substantially less training data**.
>
>     These results indicate that our hybrid design is not only efficient in terms of compute and data usage, but and effectively *maintains the original performance level* despite the significantly stricter data constraints.
>
> - **[W2] The efficiency gains are presented in a potentially misleading manner. **:
>
>   - We sincerely apologize for the lack of clarity regarding the reported “nearly 2×” throughput improvement in the abstract. This figure corresponds to a *hypothetical* configuration where **all** attention layers are replaced with RWKV layers. However, we did not actually train this variant, as it is unlikely to yield strong performance under our current resource and data constraints.
>
>     To ensure clarity and accuracy, we will revise the abstract and main text to report the *empirically validated* result:
>
>     > *“Replacing 25% of the attention layers with RWKV layers improves throughput by 20%.”*
>
>     We fully agree with the reviewer that the “up to 2×” throughput gain (corresponding to 100% RWKV replacement) is not practically viable in the current setting—especially given the notable performance drop observed at 50% replacement.
>
>     However, we hypothesize that this degradation is primarily attributable to limitations in the scale and quality of available video-language training data, rather than an inherent flaw in the hybrid architecture itself. Specifically:
>
>     - At 25% replacement, our model already achieves performance *very close* to the full attention baseline, while gaining +20% throughput.
>     - In principle, replacing an *additional* 25% layers (i.e., reaching 50%) should remain feasible—if the model receives sufficient high-quality training signal to adapt the newly introduced RWKV layers and rebalance cross-modal dynamics.
>
>     Unfortunately, the high-quality video instruction-tuning data used to train Qwen2.5-VL (on which our hybrid model is built) has not been released. As a result, we are constrained to existing open-source datasets (llava-video-178k). Under such conditions, aggressive architectural changes (e.g., 50%+ replacement) may outpace the model’s capacity to adapt, leading to suboptimal convergence.
>
>     That said, we view this not as a ceiling, but as a *data bottleneck*. With access to larger, higher-fidelity video-language corpora (akin to those used in Qwen2.5-VL).
> **[W3] The proposed method appears to be extremely brittle. The ablations reveal that the approach only works in a very specific "sweet spot": **:
>
> - As discussed in our response to [W2], we posit that replacing a larger proportion of layers inevitably diminishes the retention of the pre-trained model's inherent capabilities. Consequently, maintaining performance with extensive layer replacement would necessitate significantly higher-quality training data and greater computational resources. Unfortunately, such demands exceed our current computational constraints.

---

> ### Author Response · Authors · 2025-11-24
> **Rebuttal to Reviewer FGe7**
>
> **[W4] The 20% inference speedup (which comes with a performance trade-off, not a clear win) is achieved at the cost of a significant increase in training and architectural complexity.**:
>
> - As discussed in [W1], our scheme differs from existing hybrid architectures (e.g., VAMBA, SlowFast MLLM) by implementing more substantial architectural changes while utilizing fewer training resources. Given that Mamba-based hybrid architectures have been widely validated (e.g., Google's Nemotron-H and Tencent's Hunyuan-Turbo-S), our goal is to investigate the efficacy of RWKV-based hybridization.
>
>   Our results demonstrate significant potential:
>
>   1. **Performance:** Comparison with LongLLava shows a significant performance improvement, indicating a clear benefit beyond just inference speed.
>   2. **Training Efficiency:** Notably, we found that training the RWKV hybrid architecture offers significantly faster convergence speeds compared to Mamba hybrid architectures. This suggests that our proposed method is not only a viable alternative but potentially superior to existing Mamba-based solutions.

---

> > ### Comment · Area_Chair_yCCM · 2025-11-26
> >
> > Dear reviewer FGe7:
> >
> > Could you take a look at the response from the author and leave your feedback?
> >
> > AC.

---

### Official Review · Reviewer_C7LG · 2025-10-30

**Soundness:** 2
**Presentation:** 2
**Contribution:** 2
**Rating:** 4
**Confidence:** 3

**Summary:**

This paper introduces a hybrid RWKV-Transformer architecture to address the quadratic complexity bottleneck of Transformers in long-video understanding. Instead of reducing visual tokens (which causes information loss), the authors replace selected Transformer self-attention layers with RWKV modules, a linear RNN-style architecture, achieving pp to 2× inference speed-up without token reduction
20% throughput gain by replacing just 25% of layers. The model (based on Qwen2.5-VL-7B) outperforms efficient baselines like LongLLaVA and VAMBA while using less training data and no token compression.

**Strengths:**

1. The paper introduces the first hybrid RWKV-Transformer architecture for video MLLMs, combining the efficiency of linear RNNs with the expressiveness of Transformers.
2. The paper proposes a parameter remapping strategy that reuses pre-trained Transformer weights to initialize RWKV modules—no full retraining required. Enables fast adaptation and knowledge preservation from powerful pre-trained models like Qwen2.5-VL.

**Weaknesses:**

1. RWKV Trade-offs: RWKV’s constant-memory recurrence is efficient, but its data-dependent decay still causes slight degradation on the longest videos.
2. Training Cost. Although pre-trained weights are reused, the three-stage distillation still requires ~2.5 M video samples and 8×A800 GPUs; total GPU-hours and carbon footprint are not disclosed, limiting cost-benefit analysis.
3.  Over-hyped speed-up: the “≈2×” claim only holds if every attention layer is swapped (a configuration whose accuracy is never even reported). The promoted 25 % hybrid yields a mere 20 % throughput boost—far less than the headline suggests.

**Questions:**

1. Long-video degradation with RWKV. Figure 2 shows that 50 % layer replacement already hurts accuracy, yet the 100 % RWKV model is never evaluated. Please provide full benchmark scores for the all-RWKV configuration or present evidence that its performance is unusable; otherwise the “≈ 2 ×” speed-up lacks practical relevance.
2. Training cost and reproducibility. The three-stage distillation uses ~2.5 M videos on 8 × A800 GPUs. Please report the total GPU-hours.

---

> ### Author Response · Authors · 2025-11-24
> **Rebuttal to Reviewer C7LG**
>
> We appreciate the reviewer's insightful comments and we address the concerns below:
>
> - **[W1] RWKV Trade-offs: RWKV’s constant-memory recurrence is efficient, but its data-dependent decay still causes slight degradation on the longest videos.**:
>
>   - To explicitly mitigate this limitation, we introduce **cross-attention modules** that allow RWKV layers to attend directly to a set of global *scene tokens*.As shown in our ablation study (Table 4, Row 1 vs. Row 3), adding cross-attention yields a **+4.6% absolute gain** on the long-video subset of Video-MME —demonstrating that this design effectively compensates for RWKV’s inherent decay and significantly enhances long-range comprehension. We attribute the slight performance degradation observed on the longest videos primarily to limitations within the training data. Lacking access to the proprietary training set of Qwen-2.5-VL, we were constrained to relying on open-source video datasets. However, the duration of the videos in these public datasets is likely insufficient, resulting in inadequate training of the model's capability to process extremely long video sequences.
>
> - **[W2] Training Cost. Although pre-trained weights are reused, the three-stage distillation still requires ~2.5 M video samples and 8×A800 GPUs; total GPU-hours and carbon footprint are not disclosed, limiting cost-benefit analysis.:**
>
>   - First, we would like to clarify that our training uses approximately **1.5M** video samples, not 2.5M. The model is trained on **8×A800 GPUs for about 7 days**. Notably, our data usage is significantly **smaller** than that of VAMBA (>6M video samples) and Slow-Fast MLLM (3,467K samples). The specific training data sizes and corresponding performance comparisons are presented in the table below.
>
>     | Model                   | datasets | VideoMME |
>     | ----------------------- | -------- | -------- |
>     | Slow-fast MLLM          | 3.5M     | 60.3     |
>     | VAMBA                   | 6M       | 57.8     |
>     | Qwen-RWKV-VL(25%)(ours) | 1.5M     | 61.3     |
>
>     Both of these methods retain the original **self-attention** parameters and functionality, adding new Mamba modules or cross-attention layers on top. In contrast, our approach **fully replaces self-attention with RWKV**, introducing substantial architectural changes that would, in principle, make training more challenging. Despite this, we achieve **superior performance using even less training data**, demonstrating the efficiency and effectiveness of our design.
> **[W3] Over-hyped speed-up: the “≈2×” claim only holds if every attention layer is swapped (a configuration whose accuracy is never even reported). The promoted 25 % hybrid yields a mere 20 % throughput boost—far less than the headline suggests.:**
>
> - We sincerely apologize for the lack of clarity regarding the reported “nearly 2×” throughput improvement in the abstract. This figure corresponds to a *hypothetical* configuration where **all** attention layers are replaced with RWKV layers. However, we did not actually train this variant, as it is unlikely to yield strong performance under our current resource and data constraints.
>
>   To ensure clarity and accuracy, we will revise the abstract and main text to report the *empirically validated* result:
>
>   > *“Replacing 25% of the attention layers with RWKV layers improves throughput by 20%.”*
>
>   However, we hypothesize that this degradation is primarily attributable to limitations in the scale and quality of available video-language training data, rather than an inherent flaw in the hybrid architecture itself. Specifically:
>
>   - At 25% replacement, our model already achieves performance *very close* to the full attention baseline (e.g., −0.4 on VideoMME), while gaining +20% throughput.
>   - In principle, replacing an *additional* 25% layers (i.e., reaching 50%) should remain feasible—if the model receives sufficient high-quality training signal to adapt the newly introduced RWKV layers and rebalance cross-modal dynamics.
>
>   Unfortunately, the high-quality video instruction-tuning data used to train Qwen2.5-VL (on which our hybrid model is built) has not been released. As a result, we are constrained to existing open-source datasets (llava-video-178k). Under such conditions, aggressive architectural changes (e.g., 50%+ replacement) may outpace the model’s capacity to adapt, leading to suboptimal convergence.
>
>   That said, we view this not as a ceiling, but as a *data bottleneck*. With access to larger, higher-fidelity video-language corpora (akin to those used in Qwen2.5-VL).

---

> > ### Comment · Reviewer_C7LG · 2025-11-25
> >
> > We appreciate the authors' efforts to clarify these points; however, the rebuttal does not adequately resolve the core concerns and in some cases reinforces them.
> >  The introduction of cross-attention modules is framed as a mitigation, but this concedes rather than refutes the fundamental limitation: RWKV's data-dependent decay impairs performance on long videos. Adding auxiliary mechanisms is a workaround, not a demonstration that RWKV layers themselves can handle long-range dependencies equivalently to attention. The claim that degradation stems from "insufficient video duration" in open-source data is speculative and unsupported; performance gaps on established benchmarks like Video-MME are attributable to the architecture, not a hypothetical training data deficit.

---

> ### Author Response · Authors · 2025-11-24
> **Rebuttal to Reviewer C7LG**
>
> - **[Q1] Long-video degradation with RWKV. Figure 2 shows that 50 % layer replacement already hurts accuracy, yet the 100 % RWKV model is never evaluated. Please provide full benchmark scores for the all-RWKV configuration or present evidence that its performance is unusable; otherwise the “≈ 2 ×” speed-up lacks practical relevance.:**
>
>   - As discussed in our response to **[W3]**, training an all-RWKV variant under our current compute budget and available open-source video–language data is **unlikely to yield a sufficiently well-optimized model**. The architectural change—replacing *all* self-attention layers with RWKV—requires substantially stronger training signals and larger, higher-quality datasets to adequately adapt cross-modal representations. Unfortunately, the proprietary large-scale video instruction-tuning data used by Qwen2.5-VL (our base model) has **not** been released, and publicly available datasets (e.g., llava-video-178k) do not provide adequate coverage for such a drastic modification.
>
> - **[Q2] How does the memory consumption scale with the number of input video tokens?:**
>
>   - As described in [W2], we appreciate the reviewer’s attention to training cost and reproducibility. We would like to clarify that our training pipeline uses **approximately 1.5M** video samples (rather than 2.5M) and is trained on **8 × A800 GPUs for about 7 days**.

---

### Official Review · Reviewer_njDk · 2025-11-01

**Soundness:** 3
**Presentation:** 3
**Contribution:** 2
**Rating:** 4
**Confidence:** 4

**Summary:**

This paper addresses the significant computational bottleneck in Transformer-based Multimodal Large Language Models for long video understanding, which stems from the quadratic complexity of self-attention. The authors propose a hybrid RWKV-Transformer architecture that replaces a subset of the standard self-attention layers with Hybrid Decoder Layer. This new layer has two key components:An RWKV module that processes all visual and text tokens with linear complexity, providing the primary efficiency gain. A cross-attention module that runs in parallel. This module allows the text tokens to attend to a set of "scene tokens", which are adaptively pooled global representations of the video. This component is designed to mitigate the history decay common in RNNs by providing global context. To avoid costly retraining, the model is initialized by remapping weights from a pre-trained MLLM (Qwen2.5-VL), and a 3-stage progressive distillation strategy is used to align the new hybrid layers. The authors demonstrate that replacing 25% of the Transformer layers achieves a 20% throughput increase while matching or even slightly exceeding the baseline's performance on benchmarks like VNBench and LVBench.

**Strengths:**

1. The paper is generally well-written and well-structured. The problem is clearly stated, and the proposed architecture is explained in detail, clarifying the distinct paths for the RWKV and cross-attention modules. The 3-stage distillation process is also well-defined.

2. The paper tackles a practical problem. The "hybrid" approach and cross attention are practical and make sense.

3. The empirical validation is strong. The ablation studies in Section 5.3 are thorough and provide convincing evidence for the authors' design choices. Table 4 clearly shows the necessity of the cross-attention module to mitigate performance drops, and Table 3 rightly identifies that replacing late-stage layers is more effective than early-stage ones. The adaptive scene pooling based on similarity is also a better approach than naive uniform pooling.

**Weaknesses:**

1. The technical novelty is limited. The proposed architecture and distillation training are very similar to the discussed related work, Vamba, except that it uses RWKV instead of Mamba

2.  While the "up to 2x" throughput is highlighted, this is for a 100% replacement model that, based on the steep performance drop of the 50% model, is likely unusable. The practical, high-performance model (25% replacement) only achieves a 20% speedup.

3. The paper's design routes both text and visual tokens through the same RWKV module. An alternative, and perhaps cleaner, architecture was not explored: Apply RWKV layers only to the visual tokens (the bottleneck). Keep text tokens in standard Transformer layers. Have the text tokens cross-attend to the (now $O(N)$) RWKV-processed visual representations.

4. Lack of Memory Analysis: A critical component of "efficiency" is entirely missing from the analysis: inference memory. The primary appeal of models like RWKV is their $O(1)$ memory complexity during autoregressive generation, which contrasts with the $O(N)$ memory of a Transformer's KV-cache. Since this model is a 75% Transformer hybrid, it presumably still maintains an $O(N)$ KV-cache for the majority of its layers. This means that for "hour-long" videos, the memory bottleneck from the visual token KV-cache would still be enormous, even if the throughput is 20% faster. The paper's focus on throughput (tokens/s) alone feels incomplete.

**Questions:**

See W4. Could the authors please provide a detailed analysis of training/inference memory usage.
How does the memory consumption scale with the number of input video tokens?

---

> ### Author Response · Authors · 2025-11-24
> **Rebuttal to Reviewer njDk**
>
> We appreciate the reviewer's insightful comments and we address the concerns below:
>
> - **[W1] The technical novelty is limited**:
>
>   - We respectfully disagree with the assessment that our work lacks technical novelty. While our work is indeed inspired by VAMBA, there are several *fundamental* architectural and training differences that critically impact model efficiency, performance, and practicality:
>
>     1. Parameter-Efficient Integration:
>        VAMBA *adds* Mamba modules on top of the original attention-based architecture, increasing the parameter count from 7B to 10B. In contrast, our approach replaces a subset of attention layers with RWKV layers and transfers parameters from the original attention weights via structured initialization. This avoids parameter inflation and significantly reduces the need for extensive downstream training.
>     2. Unified Cross-Modal Modeling :
>        VAMBA processes *visual tokens exclusively with Mamba* and *text tokens exclusively with attention*, effectively decoupling the two modalities. This architectural separation severely harms multimodal reasoning: on VideoMME, its accuracy drops from 63.3% (baseline) to 57.8%. .
>     3. Training Efficiency and Data Scale:
>        VAMBA relies solely on supervised fine-tuning (SFT) with over 10 million samples. In contrast, we adopt a distillation-augmented training scheme using only 1.5 million samples .
>
> - **[W2] While the "up to 2x" throughput is highlighted, this is for a 100% replacement model that, based on the steep performance drop of the 50% model, is likely unusable. The practical, high-performance model (25% replacement) only achieves a 20% speedup.:**
>
>   - We sincerely apologize for the lack of clarity regarding the reported “nearly 2×” throughput improvement in the abstract. This figure corresponds to a *hypothetical* configuration where **all** attention layers are replaced with RWKV layers. However, we did not actually train this variant, as it is unlikely to yield strong performance under our current resource and data constraints.
>
>     To ensure clarity and accuracy, we will revise the abstract and main text to report the *empirically validated* result:
>
>     > *“Replacing 25% of the attention layers with RWKV layers improves throughput by 20%.”*
>
>     We fully agree with the reviewer that the “up to 2×” throughput gain (corresponding to 100% RWKV replacement) is not practically viable in the current setting—especially given the notable performance drop observed at 50% replacement.
>
>     However, we hypothesize that this degradation is primarily attributable to limitations in the scale and quality of available video-language training data, rather than an inherent flaw in the hybrid architecture itself. Specifically:
>
>     - At 25% replacement, our model already achieves performance *very close* to the full attention baseline, while gaining +20% throughput.
>     - In principle, replacing an *additional* 25% layers (i.e., reaching 50%) should remain feasible—if the model receives sufficient high-quality training signal to adapt the newly introduced RWKV layers and rebalance cross-modal dynamics.
>
>     Unfortunately, the high-quality video instruction-tuning data used to train Qwen2.5-VL (on which our hybrid model is built) has not been released. As a result, we are constrained to existing open-source datasets (llava-video-178k). Under such conditions, aggressive architectural changes (e.g., 50%+ replacement) may outpace the model’s capacity to adapt, leading to suboptimal convergence.
>
>     That said, we view this not as a ceiling, but as a *data bottleneck*. With access to larger, higher-fidelity video-language corpora (akin to those used in Qwen2.5-VL).
> - **[W3] The paper's design routes both text and visual tokens through the same RWKV module. An alternative, and perhaps cleaner, architecture was not explored: Apply RWKV layers only to the visual tokens (the bottleneck). Keep text tokens in standard Transformer layers. Have the text tokens cross-attend to the (now ) RWKV-processed visual representations.:**
>
>   - As discussed in [W1], VAMBA has indeed explored this modality-separate approach—processing visual tokens with Mamba and textual tokens with attention. However, their results show a significant performance drop (e.g., from 63.3% to 57.8% on VideoMME), indicating that decoupling visual and textual representations harms cross-modal alignment and reasoning.
>
>     Given this empirical evidence, we intentionally avoided such a split design in our work. Instead, we adopt a *unified hybrid backbone* where both modalities are jointly modeled within the same layers.

---

> ### Author Response · Authors · 2025-11-24
> **Rebuttal to Reviewer njDk**
>
> - **[W4] Lack of Memory Analysis.:**
>
>   - We sincerely thank the reviewer for raising this important point. The reviewer is absolutely correct that inference memory footprint—especially KV-cache usage during autoregressive generation—is a crucial dimension of efficiency, particularly for long-context video understanding.
>
>     In our current hybrid design (75% attention + 25% RWKV), we do not reduce model parameter count, nor do we apply any token compression. Consequently, the peak GPU memory consumption remains comparable to the original Qwen2.5-VL—approximately 30 GB for hour-long videos , dominated by the KV-cache of the retained attention layers.
>
>     That said, the primary goal of this work is acceleration within the feasibility envelope of the original model:If the base model can run inference on a given hardware setup, our hybrid variant can also run—while delivering higher throughput.
>
>     Nevertheless, we fully agree that memory efficiency is a vital next step. Our hybrid architecture is *orthogonal* to—and can be naturally combined with—token compression techniques.For example, compressing visual tokens *before* they enter the hybrid backbone would directly reduce the KV-cache burden across all attention layers. We view this as a highly promising direction for future work.
>
> - **[Q1] How does the memory consumption scale with the number of input video tokens?:**
>
>   - As described in [W4], our model's GPU memory consumption scale for both training and inference remains almost the same as the original model. It is not particularly informative to directly compare training memory, however, because our distillation method necessitates loading the weights of a teacher model as well. During inference, the memory consumption begins at 18GB upon loading the model and can increase to as much as 30GB as more video frames are input, as long as the model's maximum input length is not surpassed.

---

> > ### Comment · Area_Chair_yCCM · 2025-11-26
> >
> > Dear reviewer njDk:
> >
> > Could you take a look at the response from the author and leave your feedback?
> >
> > AC.

---

### Official Review · Reviewer_NcrW · 2025-11-01

**Soundness:** 2
**Presentation:** 2
**Contribution:** 3
**Rating:** 4
**Confidence:** 5

**Summary:**

This paper proposes a hybrid RWKV-Transformer model for efficient long-video understanding. It focuses on distilling a retrained Transformer model into a linear RNN-based model, achieving 2× higher throughput. In addition, the authors employ cross-attention to help the RWKV capture global context, mitigating its inherent history-decay issue

**Strengths:**

* Motivation is good: focus on archeiture design of long vide-LLMs
* method is reasonable, efficiently replace the complex attention block with linear attention block
* the cross-attention is effecively to mitigate  RWKV's inherent history-decay issue of RWKV

**Weaknesses:**

* Some clarifications are not clear.
  For example, in the abstract, the authors mention that the throughput increases by nearly 2×.
  Under what setting is this achieved? Does it maintain the same performance?

* Insufficient baselines:
  * Throughput comparison with other baselines such as VAMBA and SlowFast-MLLM.
    It would be better to include a throughput-versus-performance curve to compare your model with these baselines.
  * Ablation using the same fine-tuning data on the original architecture.
  * What is the performance when replacing the causal attention with RWKV without any additional training?

**Questions:**

Is the difference between Stage 2 and Stage 1 that Stage 2 includes MLP training and uses different data?

Also, why did the authors specifically choose RWKV? There are many types of linear attention mechanisms to choose from — I would like to know the authors’ reasoning behind this choice.

---

> ### Author Response · Authors · 2025-11-24
> **Rebuttal to Reviewer NcrW**
>
> We appreciate the reviewer's insightful comments and we address the concerns below:
>
> - **[W1] Some clarifications are not clear.**:
>
>   - We sincerely apologize for the lack of clarity regarding the reported “nearly 2×” throughput improvement in the abstract. This figure corresponds to a *hypothetical* configuration where **all** attention layers are replaced with RWKV layers. However, we did not actually train this variant, as it is unlikely to yield strong performance under our current resource and data constraints.
>
>     To ensure clarity and accuracy, we will revise the abstract and main text to report the *empirically validated* result:
>
>     > *“Replacing 25% of the attention layers with RWKV layers improves throughput by 20%.”*
>
>     We fully agree with the reviewer that the “up to 2×” throughput gain (corresponding to 100% RWKV replacement) is not practically viable in the current setting—especially given the notable performance drop observed at 50% replacement.
>
>     However, we hypothesize that this degradation is primarily attributable to limitations in the scale and quality of available video-language training data, rather than an inherent flaw in the hybrid architecture itself. Specifically:
>
>     - At 25% replacement, our model already achieves performance *very close* to the full attention baseline, while gaining +20% throughput.
>     - In principle, replacing an *additional* 25% layers (i.e., reaching 50%) should remain feasible—if the model receives sufficient high-quality training signal to adapt the newly introduced RWKV layers and rebalance cross-modal dynamics.
>
>     Unfortunately, the high-quality video instruction-tuning data used to train Qwen2.5-VL (on which our hybrid model is built) has not been released. As a result, we are constrained to existing open-source datasets (llava-video-178k). Under such conditions, aggressive architectural changes (e.g., 50%+ replacement) may outpace the model’s capacity to adapt, leading to suboptimal convergence.
>
>     That said, we view this not as a ceiling, but as a *data bottleneck*. With access to larger, higher-fidelity video-language corpora (akin to those used in Qwen2.5-VL).

---

> ### Author Response · Authors · 2025-11-24
> **Rebuttal to Reviewer NcrW**
>
> **[W2] Insufficient baselines**:
>
> - **Throughput comparison with other baselines such as VAMBA and SlowFast-MLLM.** 、
>
>   We appreciate the reviewer’s observation regarding throughput comparisons across different models. Indeed, due to significant variations in input video resolution, frame count, and architectural design, direct throughput comparisons between models may not always be meaningful or fair.Nevertheless, we attempted to conduct a comparative experiment. Since VAMBA performed poorly on the significant VideoMME benchmark, we chose to only compare the throughput of SlowFast-MLLM with our hybrid model, in which 25% of the components have been replaced.We tested the throughput by inputting 256 frames at a resolution of 576x576, and the results are shown in the table below.
>
>   | Model                       | Prefill Time | throughput    |
>   | --------------------------- | ------------ | ------------- |
>   | Slow-fast MLLM              | 2.8835s      | 26.93Tokens/s |
>   | **Qwen-RWKV-VL(25%)(ours)** | 0.5589s      | 36.47Tokens/s |
>
>   As can be seen from the results, our hybrid model significantly outperforms SlowFast-MLLM in both prefill time and throughput.
>
> - **Ablation using the same fine-tuning data on the original architecture.**
>
>   We fine-tuned the original Qwen2.5-VL-7B model using video data from all distillation stages and evaluated it across various video benchmarks. The results, compared with the original model, are presented below:
>
>   | Model                    | VNBench  | VideoMME | MLVU     | LongVideoBench | LVBench  |
>   | ------------------------ | -------- | -------- | -------- | -------------- | -------- |
>   | Qwen2.5-VL-7B(oringinal) | 73.4     | **63.8** | **70.4** | **49.5**       | 45.3     |
>   | Qwen2.5-VL-7B(finetuned) | 69.9     | 58.4     | 63.9     | 43.7           | 40.5     |
>   | Qwen-RWKV-VL(25%)(ours)  | **74.0** | 61.3     | 68.0     | 47.8           | **46.8** |
>
>   It is evident that the performance of the fine-tuned model has declined significantly. This further underscores the insufficient quality of existing open-source video datasets, which restricts the capabilities of the models we train，we thus hypothesize that with access to higher-fidelity data (e.g., proprietary instruction data akin to that used in Qwen2-VL), our architecture could not only match but potentially surpass the original model, while simultaneously offering improved inference efficiency.
>
> - **What is the performance when replacing the causal attention with RWKV without any additional training?**
>
>   If we directly replace attention layers with RWKV *without any training*, although the majority of RWKV parameters are initialized via parameter conversion from the original attention weights, a small subset remains randomly initialized. We therefore consider such an untrained variant largely uninformative. Nevertheless, for completeness, we conducted this ablation; the results on VideoMME are as follows:
>
>   | Model                                     | short | medium | long | avg  |
>   | ----------------------------------------- | ----- | ------ | ---- | ---- |
>   | Qwen-RWKV-VL(25%)(ours)(without training) | 43.9  | 31.2   | 23.6 | 32.7 |
>
>   The model performs very poorly—nearly at random-guessing level—confirming that minimal adaptation is essential for functional deployment.
> - **[Q1]Is the difference between Stage 2 and Stage 1 that Stage 2 includes MLP training and uses different data?**:
>
>   - Yes. As our experiments show, jointly training the newly added RWKV layers (with partially randomly initialized parameters) and the original MLP layers from the beginning tends to **disrupt the pre-trained knowledge in the MLP**, leading to unstable optimization. To avoid this, we decouple the training into two stages.
>
>     Moreover, we find that **curriculum-style progression**—first training on short videos (Stage 1) and then moving to medium- and long-duration videos (Stage 2)—enables more stable and consistent performance improvement. In contrast, mixing video lengths from the start causes the model to quickly plateau, with little further gain observed thereafter.
>
> - **[Q2]Also, why did the authors specifically choose RWKV?**:
>
>   - First, hybrid architectures based on Mamba have already been validated in a range of large-scale systems (e.g., Google’s Nemotron-H and Tencent’s HunYuan-Turbo-S). Motivated by this, we aim to explore whether alternative linear-attention models—such as RWKV—can achieve comparable or even superior performance.
>
>     In our experiments, we observe that the RWKV-based hybrid architecture exhibits **faster training speed** and **quicker convergence** compared to its Mamba-based counterpart. Moreover, RWKV-based *language-only* models have already demonstrated strong text generation capabilities. Building on this, we seek to investigate whether RWKV can also support effective multimodal—particularly video—understanding when integrated into a vision-language foundation model.

---

> > ### Comment · Area_Chair_yCCM · 2025-11-26
> >
> > Dear reviewer NcrW:
> >
> > Could you take a look at the response from the author and leave your feedback?
> >
> > AC.

---

### Meta-Review · Area_Chair_v9ms · 2025-12-29

**Summary:**

The main weaknesses of the paper are focused around limited novelty over prior hybrid architectures, over‑stated efficiency claims, brittleness of the method, and incomplete efficiency analysis. Reviewer NcrW sees “insufficient baselines,” including missing throughput comparisons against models like VAMBA and SlowFast‑MLLM, and asks for ablations using the same fine‑tuning data on the original architecture and for performance when replacing attention with RWKV without training; they also ask why RWKV is chosen over other linear mechanisms. Reviewer njDk argues that “the technical novelty is limited,” stating that the approach is “very similar to the discussed related work, VAMBA, except that it uses RWKV instead of Mamba,” and notes that the advertised “up to 2x” throughput comes from a 100% replacement model whose accuracy is never reported, while the practical 25% model achieves only 20% speedup (which is still significant); they also highlight that both text and visual tokens go through the same RWKV module, and that there is a “lack of Memory Analysis,” since the hybrid still has an O(N) KV cache for the remaining Transformer layers. Reviewer C7LG calls out “over-hyped speed-up,” noting that the ≈2× claim is tied to a configuration that is not evaluated and that the 25% variant only gives a 20% gain; they also flag that RWKV’s decay still leads to degradation on long videos, that training cost and GPU‑hours are not reported, and that the RWKV trade‑offs are not fully addressed. Reviewer FGe7 writes that “the central claim that the 25% model ‘match[es] its performance’ … is an overstatement,” since performance drops on Video‑MME and MLVU, and criticizes the method as “extremely brittle,” working only at a specific replacement ratio and failing at 50%; they conclude that the efficiency gains are modest and come “at the cost of a significant increase in training and architectural complexity.”

The AC recommends rejection following reviewer recommendation because, despite interesting engineering and solid empirical work, the contribution does not clearly rise above existing hybrid designs and the efficiency–performance trade‑off is less favorable than the paper suggests. Reviewer NcrW and Reviewer njDk both converge on “marginally below acceptance,” with concerns about limited novelty over VAMBA‑like hybrids, incomplete efficiency characterization (no real memory gains), and missing or only partially comparable baselines. Reviewer C7LG and Reviewer FGe7 are more critical: they argue that the “≈2×” speed‑up claim is misleading since the only trained and usable configuration provides a 20% throughput gain (which the AC thinks is still significant) while underperforming the Qwen2.5‑VL baseline on key benchmarks, and they view the method as brittle, working only at a carefully tuned 25% replacement and failing as the replacement ratio increases. The rebuttal is careful and adds useful details on throughput, training cost, and baselines, and the authors agree to tone down over‑claims; however, it does not change the underlying facts that the realized acceleration is more modest, that the architecture is close in spirit to prior hybrid systems, and that scalability to more aggressive replacement remains a speculation. With three reviewers at 4 and one at 2, and with the main reservations about novelty, robustness, and claim strength still present after rebuttal, the AC does not find sufficient grounds to recommend acceptance.

On balance, the AC sees no basis to overturn the reviewer suggestions (and their consensus towards rejection) and recommends rejection. The AC highly recommends the authors to address the concerns of the reviewers and take into account their suggestions of improvement when preparing a revised version.

**Reviewer Concerns:**

The rebuttal addresses many specific technical questions but does not fully resolve the core conceptual and empirical concerns. To Reviewer NcrW, the authors clarify that the “nearly 2x” figure is hypothetical and commit to revising it to “replacing 25% of the attention layers … improves throughput by 20%,” and they provide a throughput comparison against SlowFast‑MLLM, as well as an ablation where the original Qwen2.5‑VL is fine‑tuned on the same video data (which underperforms both the original baseline and the hybrid), and results for a variant with attention replaced by RWKV without training (which performs very poorly). They also explain that Stage 2 differs from Stage 1 in including MLP training and longer videos, and justify RWKV by pointing to faster convergence than Mamba and strong RWKV language models. To Reviewer njDk and Reviewer C7LG, they argue that their approach is more parameter‑efficient than VAMBA, that VAMBA’s modality‑separated design harms multimodal reasoning, and that their data usage (about 1.5M videos, 8xA800 for ~7 days) is lower than VAMBA and SlowFast‑MLLM; they also state that memory usage remains comparable to Qwen2.5‑VL(~18–30GB) and position their work as orthogonal to token compression. They repeatedly promise to correct the “2x” wording and attribute the failure of higher replacement ratios to data limitations rather than the architecture. Reviewer C7LG responds that this does not resolve the underlying limitation and views the explanation as speculative. To Reviewer FGe7, they reiterate that performance is “comparable”given less data and argue that, compared to other hybrid architectures, their stronger architectural change with fewer resources is meaningful, but they do not change the basic picture of performance drops on some key benchmarks and brittleness beyond 25% replacement. Overall, the rebuttal improves clarity, adds baselines and training‑cost details, and corrects the strongest claim, but it does not substantially alter the limited novelty over VAMBA‑style hybrids, the modest realized speedup, or the delicate nature of the replacement schedule.

**Reviewer Scores:**

Given that reviewers could not update their scores after the rebuttal, we can only speculate about any potental shifts in scores. Reviewer NcrW, at rating 4 may be somewhat reassured by the added throughput comparison and ablation results, but their criticisms about baselines and clarity are only partially alleviated. Reviewer njDk, also at 4, sees technical novelty as limited and raises the missing memory analysis; the authors’ acknowledgment that memory is unchanged and that their goal is primarily acceleration probably would not increase this rating. Reviewer C7LG, at 4, explicitly states post‑rebuttal that their core concerns “are not adequately resolved” and that the data‑bottleneck explanation is speculative, indicating they would not raise their score. Reviewer FGe7, at “2 reject” emphasizes overstatement of claims, brittleness, and complexity; the corrections in wording and the added context around data limitations are unlikely to fully change this assessment.

---

### Decision · Program_Chairs · 2026-01-26

Reject